# Hospital-Acquired Dysmagnesemia and In-Hospital Mortality

**DOI:** 10.3390/medsci8030037

**Published:** 2020-09-01

**Authors:** Wisit Cheungpasitporn, Charat Thongprayoon, Api Chewcharat, Tananchai Petnak, Michael A. Mao, Paul W. Davis, Tarun Bathini, Saraschandra Vallabhajosyula, Fawad Qureshi, Stephen B. Erickson

**Affiliations:** 1Division of Nephrology and Hypertension, Department of Medicine, Mayo Clinic, Rochester, MN 55905, USA; api.che@hotmail.com (A.C.); Qureshi.Fawad@mayo.edu (F.Q.); erickson.stephen@mayo.edu (S.B.E.); 2Division of Nephrology, Department of Internal Medicine, University of Mississippi Medical Center, Jackson, MS 39216, USA; pwdavis@umc.edu; 3Division of Pulmonary and Pulmonary Critical Care Medicine, Department of Medicine, Faculty of Medicine, Ramathibodi Hospital, Mahidol University, Bangkok 10400, Thailand; petnak@yahoo.com; 4Division of Pulmonary and Critical Care Medicine, Department of Medicine, Mayo Clinic, Rochester, MN 55905, USA; 5Division of Nephrology and Hypertension, Mayo Clinic, Jacksonville, FL 32224, USA; mao.michael@mayo.edu; 6Department of Internal Medicine, University of Arizona, Tucson, AZ 85721, USA; tarunjacobb@gmail.com; 7Section of Interventional Cardiology, Division of Cardiovascular Medicine, Department of Medicine, Emory University School of Medicine, Atlanta, GA 30322, USA; svalla4@emory.edu

**Keywords:** hypomagnesemia, hypermagnesemia, magnesium, electrolytes, mortality, hospitalization, nephrology, internal medicine

## Abstract

**Background and Objectives:** This study aimed to report the incidence of hospital-acquired dysmagnesemia and its association with in-hospital mortality in adult general hospitalized patients. **Materials and Methods:** We studied 26,020 adult hospitalized patients from 2009 to 2013 who had normal admission serum magnesium levels and at least two serum magnesium measurements during hospitalization. The normal range of serum magnesium was 1.7–2.3 mg/dL. We categorized in-hospital serum magnesium levels based on the occurrence of hospital-acquired hypomagnesemia and/or hypermagnesemia. We assessed the association between hospital-acquired dysmagnesemia and in-hospital mortality using multivariable logistic regression. **Results:** 28% of patients developed hospital-acquired dysmagnesemia. Fifteen per cent had hospital-acquired hypomagnesemia only, 10% had hospital-acquired hypermagnesemia only, and 3% had both hospital-acquired hypomagnesemia and hypermagnesemia. Compared with patients with persistently normal serum magnesium levels in hospital, those with hospital-acquired hypomagnesemia only (OR 1.77; *p* < 0.001), hospital-acquired hypermagnesemia only (OR 2.31; *p* < 0.001), and both hospital-acquired hypomagnesemia and hypermagnesemia (OR 2.14; *p* < 0.001) were significantly associated with higher in-hospital mortality. **Conclusions:** Hospital-acquired dysmagnesemia affected approximately one-fourth of hospitalized patients. Hospital-acquired hypomagnesemia and hypermagnesemia were significantly associated with increased in-hospital mortality.

## 1. Introduction

Magnesium is a divalent cation that plays many pivotal roles in the human body. These include muscle contraction, nerve conduction, blood glucose regulation, and blood pressure control [1,2,3,4,5,6]. The body consequently needs to closely maintain serum magnesium levels within the physiologic range of 1.7–2.4 mg/dL (0.7–1.0 mmol/L) [7]. This regulation is facilitated mainly by intestinal absorption and renal excretion [8].

Serum magnesium derangements have been associated with serious complications, such as acute kidney injury [9], septic shock [10], cardiac arrhythmias [11], respiratory failure [12], in-hospital mortality, and long-term mortality [9,13]. Previous literature has demonstrated an association between serum magnesium derangements at hospital admission and in-hospital mortality [9,14]. Assessing hospital-acquired serum magnesium derangements may provide additional prognostic information on patient outcomes and suggest novel strategies for monitoring and managing serum magnesium levels. Additionally, the incidence of hospital-acquired hypomagnesemia and hypermagnesemia among general hospitalized patients has not been well-described.

We hypothesized that hospital-acquired serum magnesium derangements are common and associated with higher in-hospital mortality. Thus, we conducted this retrospective cohort study in general hospitalized patients to report the incidence of hospital-acquired dysmagnesemia and their association with in-hospital mortality.

## 2. Materials and Methods

### 2.1. Study Population

The Mayo Clinic Institutional Review Board reviewed and approved this study (IRB number: 15-000024; Approval Date: 4 February 2015). The need for informed consent was waived because this was a minimal risk study. However, all included patients were required to provide research authorization for their health information. We identified all adult patients admitted to Mayo Clinic Hospital, Rochester, from January 2009 to December 2013. To examine the incidence and impact of hospital-acquired serum magnesium derangements, we included patients with (1) an admission serum magnesium level in the normal range of 1.7 to 2.3 mg/dL and (2) at least two serum magnesium measurements during hospitalization.

### 2.2. Definition of Hospital-Acquired Hypomagnesemia and Hypermagnesemia

We extracted clinical information and laboratory data from our institutional electronic medical record system. We defined the normal range of serum magnesium as 1.7–2.3 mg/dL based on the results from our previous study [9]. Each patient’s magnesium levels were reviewed to determine the highest and lowest in-hospital values. We identified hospital-acquired hypomagnesemia when the lowest serum magnesium level during hospital stay was less than 1.7 mg/dL, and hospital-acquired hypermagnesemia when the highest serum magnesium level during hospital stay was more than 2.3 mg/dL. We categorized in-hospital magnesium levels based on the occurrence of hospital-acquired dysmagnesemia into four groups: (1) persistently normal in-hospital magnesium levels, (2) hospital-acquired hypomagnesemia only, (3) hospital-acquired hypermagnesemia only, and (4) both hospital-acquired hypomagnesemia and hypermagnesemia.

### 2.3. Covariates

We categorized principle diagnoses based on the International Classification of Diseases, 9th Revision codes. We abstracted comorbid conditions using a previously validated data abstraction algorithm, and we calculated the Charlson Comorbidity Index to assess burden of comorbid conditions at the time of admission [15]. We estimated glomerular filtration rate (eGFR) based on age, sex, race, and admission serum creatinine using the Chronic Kidney Disease Epidemiology Collaboration (CKD-EPI) equation [16]. We defined acute kidney injury (AKI) as an increase in serum creatinine of ≥0.3 mg/dL or ≥1.5 times baseline value [17].

### 2.4. Outcomes

The primary outcome was in-hospital mortality. Death status was determined from the institutional database.

### 2.5. Statistical Analysis

We reported continuous variables either as mean ± standard deviation (SD) or median with interquartile range (IQR), as appropriate, and compared the difference between in-hospital serum magnesium groups using analysis of variance (ANOVA). We reported categorical variables as number with percentage and compared the differences between in-hospital serum magnesium groups using Chi-squared test. We performed logistic regression analysis to assess the association of hospital-acquired serum magnesium derangements, including hospital-acquired hypomagnesemia only, hospital-acquired hypermagnesemia only, and both hospital-acquired hypomagnesemia and hypermagnesemia with in-hospital mortality, compared with the persistently normal serum magnesium group. Analysis for the association was adjusted for age, sex, race, principle diagnosis, comorbidities, eGFR, AKI, intensive care unit admission, number of in-hospital serum magnesium measurements, length of hospital stay, and admission serum magnesium. We regarded 2-tailed *p* value < 0.05 as statistically significant. We used JMP statistical software (Version 10; SAS Institute Inc, Cary, NC, USA) for all analyses.

## 3. Result

### 3.1. Incidence of Hospital-Acquired Hypomagnesemia and Hypermagnesemia

A total of 26,020 adult hospitalized patients with normal admission serum magnesium levels were included in the analysis. Fifty-six percent of enrolled patients were male. The mean age was 63 ± 17 years. The median number of in-hospital serum magnesium measurements was 3 (2–6) and length of hospital stay was 5 (3–9) days. Of 26,020 hospitalized patients, 4723 (18%) and 3242 (12%) developed hospital-acquired hypomagnesemia and hypermagnesemia, respectively. A total of 18,802 (72%) had persistently normal in-hospital serum magnesium levels, 3976 (15%) had hospital-acquired hypomagnesemia only, 2495 (10%) had hospital-acquired hypermagnesemia only, and 747 (3%) had both hospital-acquired hypomagnesemia and hypermagnesemia. Patient clinical characteristics based on in-hospital serum magnesium groups are shown in Table 1. 

### 3.2. Association of Hospital-Acquired Hypomagnesemia and Hypermagnesemia with In-Hospital Mortality

Associated mortality was compared among the aforementioned groups. Of patients with hospital-acquired hypomagnesemia, 5.3% died in-hospital compared to 2.1% of patients without hospital-acquired hypomagnesemia (*p* < 0.001) (Table 2). Multivariable analysis showed that hospital-acquired hypomagnesemia was significantly associated with higher in-hospital mortality with odds ratio of 1.47 (95% CI 1.21–1.79).

Similarly, 7.6% of patients with hospital-acquired hypermagnesemia died in-hospital compared to 1.9% of patients without hospital-acquired hypermagnesemia (Table 2). Multivariable analysis showed hospital-acquired hypermagnesemia was significantly associated with higher in-hospital mortality with odds ratio of 1.91 (95% CI 1.57–2.31).

In-hospital mortality was 4.0% in patients with hospital-acquired hypomagnesemia only, 6.3% in patients with hospital-acquired hypermagnesemia only, and 12.1% in patients with both hypomagnesemia and hypermagnesemia, as compared to 1.5% in patients with persistently normal serum in-hospital magnesium levels (Table 2). Multivariable analysis showed hypomagnesemia only, hypermagnesemia only, and both hypomagnesemia and hypermagnesemia were significantly associated with increased in-hospital mortality with adjusted odds ratios of 1.77 (95% CI 1.42–2.21), 2.31 (95% CI 1.85–2.87), and 2.14 (95% CI 1.52–3.02), respectively.

The association persisted after patients with in-hospital pH < 7.35 or >7.45 were excluded from analysis. In this sensitivity analysis, hypomagnesemia only, hypermagnesemia only, and both hypomagnesemia and hypermagnesemia were significantly associated with increased in-hospital mortality with adjusted odds ratios of 1.58 (95% CI 1.07–2.33), 3.18 (95% CI 2.18–4.64), and 2.56 (1.04–6.32), respectively.

## 4. Discussion

Our study showed that the incidence of hospital-acquired hypomagnesemia only was 15%, hospital-acquired hypermagnesemia only was 10%, and both hospital-acquired hypomagnesemia and hypermagnesemia was 3%. Patients who developed hospital-acquired hypomagnesemia and hypermagnesemia were associated with higher in-hospital mortality compared to patients who did not. The association also persisted in the sensitivity analysis excluding patients with in-hospital pH <7.35 or >7.45.

Hospital-acquired dysmagnesemia is not uncommon. Previous literature reported hypomagnesemia rates ranging from 9–65% and hypermagnesemia rates ranging from 5.7–23.6% [18,19,20,21,22]. Higher incidences of both hypomagnesemia and hypermagnesemia were observed in the intensive care unit setting. Our study showed that more than one fourth of general hospitalized patients developed new serum magnesium derangements in hospital. Our data suggested that serum magnesium derangements often occur despite their association with higher in-hospital mortality and several serious complications such as cardiac arrhythmia, respiratory failure, acute kidney injury [9,10,11,12,14,23,24].

Hospital-acquired hypomagnesemia is attributed to (1) decreased intake or absorption by the gastrointestinal tract; (2) increased renal loss, due to effects of medications, such as amphotericin, chemotherapeutic agents, diuretics, or other tubular toxins; and (3) redistribution triggered by severe illnesses, such as acute pancreatitis, refeeding syndrome or cardiopulmonary surgeries [1,25]. Hypomagnesemia can lead to cardiac arrhythmias by impairing activity of Na/K ATPase and decreasing the negative resting membrane potential [26]. Hypomagnesemia is associated with higher cardiovascular disease as magnesium has a protective role for vascular calcification. Magnesium might bind with phosphate, thereby interfere with calcium phosphate deposition in vessel wall [27,28]. Additionally, hypomagnesemia can precipitate acute ischemia via endothelial dysfunction and hypercoagulability [11]. It is also associated with respiratory failure, especially in patients with asthma or chronic obstructive pulmonary disease [29,30]. These complications can all lead to increased mortality.

In contrast, hospital-acquired hypermagnesemia is commonly caused by excessive administration of magnesium-containing medications. Reported clinical cases include administration of antacids or intravenous magnesium infusions in the elderly or in those with impaired renal function [31]. Hypermagnesemia could result in neuromuscular dysfunction, respiratory depression, hypotonia, areflexia, and coma. Moreover, hypermagnesemia has been associated with malignant cardiac arrhythmias, such as complete heart block, asystole, and cardiac arrest [32,33]. Hospital-acquired hypermagnesemia is more commonly found in patients with acute kidney injury or admitted to the intensive care unit. Hence, the higher crude in-hospital mortality among hypermagnesemia patients in our study may be partially explained by their higher burden of acute and comorbid illnesses. However, the association between hospital-acquired hypermagnesemia and higher in-hospital mortality remained significant after extensive adjustment for principal diagnosis, ICU admission, comorbidities, kidney function, in-hospital AKI, and number of in-hospital serum magnesium measurements.

There are limitations to our study. Firstly, our investigation was a retrospective study conducted at a single-center hospital that predominantly composed of white patients. Consequently, causal relationships and generalizability of our findings are limited. Secondly, despite extensive adjustments for potential confounders, we did not have data on the causes of serum magnesium derangements, severity of hospital illnesses, acid-base status at the time of serum magnesium measurement, treatment of serum magnesium derangements including oral or intravenous magnesium supplements, or other medications that may have affected serum magnesium levels (e.g., diuretics, antibiotics, proton pump inhibitors, chemotherapy) [34]. This data was unavailable in our database, and therefore, our analysis may be confounded by these unadjusted confounders.

## 5. Conclusions

In conclusion, hospital-acquired dysmagnesemia affected approximately one-fourth of hospitalized patients. Hospital-acquired hypomagnesemia and hypermagnesemia were significantly associated with increased in-hospital mortality. In-hospital serum magnesium levels should be periodically monitored even if patients are admitted with initially normal serum magnesium levels.

## Figures and Tables

**Table 1 medsci-08-00037-t001:** Clinical characteristics of study patients.

Variables	All	Serum Magnesium during Hospitalization				
		Normal	Hypomagnesemia Only	Hypermagnesemia Only	Both Hypo- and Hypermagnesemia	*p*-Value
N	26020	18802	3976	2495	747	
Age (year)	63 ± 17	64 ± 17	62 ± 17	63 ± 16	62 ± 16	<0.001
Male sex	14,672 (56)	10,853 (58)	1887 (47)	1572 (63)	360 (48)	<0.001
Caucasian	24,017 (92)	17,435 (93)	3593 (90)	2316 (93)	673 (90)	<0.001
Principal diagnosis						
-Cardiovascular	7537 (29)	5847 (31)	675 (17)	815 (33)	200 (27)	<0.001
-Hematology/oncology	4103 (16)	2877 (15)	666 (17)	439 (18)	121 (16)	
-Infectious disease	1090 (4)	627 (3)	266 (7)	128 (5)	69 (9)	
-Endocrine/metabolic	947 (4)	660 (4)	211 (5)	53 (2)	23 (3)	
-Respiratory	1270 (5)	870 (5)	168 (4)	186 (7)	46 (6)	
-Gastrointestinal	3438 (13)	2365 (13)	722 (18)	241 (10)	110 (15)	
-Genitourinary	796 (3)	495 (3)	234 (6)	50 (2)	17 (2)	
-Injury and poisoning	3976 (15)	2810 (15)	639 (16)	406 (16)	121 (16)	
-Other	2863 (11)	2251 (12)	395 (10)	177 (7)	40 (5)	
Charlson comorbidity score	2.1 ± 2.5	2.0 ± 2.5	2.4 ± 2.6	2.1 ± 2.4	2.0 ± 2.3	<0.001
Comorbidity						
-Coronary artery disease	6083 (23)	4415 (23)	863 (22)	627 (25)	178 (24)	0.01
-Congestive heart failure	2337 (9)	1646 (9)	330 (8)	284 (11)	77 (10)	<0.001
-Peripheral artery disease	1104 (4)	765 (4)	193 (5)	106 (4)	40 (5)	0.06
-Stroke	2288 (9)	1628 (9)	370 (9)	222 (9)	68 (9)	0.6
-Diabetes mellitus	5728 (22)	3962 (21)	1038 (26)	548 (22)	180 (24)	<0.001
-COPD	2728 (10)	1867 (10)	427 (11)	318 (13)	116 (16)	<0.001
-Cirrhosis	845 (3)	495 (3)	234 (6)	78 (3)	38 (5)	<0.001
eGFR (mL/min/1.73 m^2^)	74 ± 31	76 ± 29	67 ± 37	71 ± 29	64 ± 35	<0.001
Acute kidney injury	8722 (34)	5186 (28)	1854 (47)	1183 (47)	499 (67)	<0.001
ICU admission	11354 (44)	7285 (39)	1918 (48)	1536 (62)	615 (82)	<0.001
Number of serum magnesium measurement in hospital	3 (2–6)	3 (2–4)	6 (4–10)	6 (4–11)	19 (10–35)	<0.001
Length of hospital stay (day)	5 (3–9)	5 (3–7)	8 (5–13)	9 (5–15)	20 (11–37)	<0.001
Admission serum magnesium (mg/dL)	1.9 ± 0.2	1.9 ± 0.2	1.9 ± 0.2	2.0 ± 0.2	1.9 ± 0.2	<0.001
Lowest serum magnesium (mg/dL)	1.8 ± 0.2	1.9 ± 0.1	1.5 ± 0.1	1.9 ± 0.2	1.5 ± 0.1	<0.001
Highest serum magnesium (mg/dL)	2.1 ± 0.2	2.1 ± 0.1	2.0 ± 0.2	2.5 ± 0.2	2.6 ± 0.4	<0.001

Continuous data are presented as mean ± SD or median (IQR); categorical data are presented as count (%).

**Table 2 medsci-08-00037-t002:** Association between hospital-acquired serum magnesium derangements and in-hospital mortality.

Serum Magnesium during Hospitalization	N	In-Hospital Mortality	Univariable Analysis		Multivariable Analysis	
			OR (95% CI)	*p*	Adjusted OR (95% CI)	*p*
Hospital-acquired hypomagnesemia						
No	21,297	438 (2.1)	1 (ref)	-	1 (ref)	-
Yes	4723	248 (5.3)	2.64 (2.25–3.09)	<0.001	1.47 (1.21–1.79)	<0.001
Hospital-acquired hypermagnesemia						
No	22,778	439 (1.9)	1 (ref)	-	1 (ref)	-
Yes	3242	247 (7.6)	4.20 (3.57–4.93)	<0.001	1.91 (1.57–2.31)	<0.001
Groups						
Normal	18,802	281 (1.5)	1 (ref)	-	1 (ref)	-
Hypomagnesemia only	3976	158 (4.0)	2.73 (2.24–3.32)	<0.001	1.77 (1.42–2.21)	<0.001
Hypermagnesemia only	2495	157 (6.3)	4.43 (3.62–5.41)	<0.001	2.31 (1.85–2.87)	<0.001
Both hypo- and hypermagnesemia	747	90 (12.1)	9.03 (7.03–11.59)	<0.001	2.14 (1.52–3.02)	<0.001

Adjusted for age, sex, race, principal diagnosis, Charlson comorbidity score, coronary artery disease, congestive heart failure, peripheral vascular disease, stroke, diabetes mellitus, chronic obstructive pulmonary disease, cirrhosis, eGFR, acute kidney injury, ICU admission, the number of serum magnesium measurement, length of hospital stay, admission serum magnesium.

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
