# Peer review of "Hospital-Acquired Dysmagnesemia and In-Hospital Mortality"

_medsci, 2020, doi:10.3390/medsci8030037_

Round 1
Reviewer 1 Report
I liked this paper a lot.
One suggestion for development of the results and discussion sections please:
Hypermagnaesemia is associated with metabolic acidosis. Patients who are initially hypermagnaesemic, may become hypomagnesaemic following correction of acidosis (similar to K+ and DKA). The inverse is also true and may carry more adverse prognostic significance. Therefore, hypermagnesemia in the context of metabolic acidosis may mask a magnesium deficient state. Hence, these fluxes may reflect the severity of metabolic derangement and explain the higher mortality among these patients.
i.e. serum Mg should be interpreted in the context of acid base balance as expressed in pH or H+.
In order for this to be a comprehensive retrospective observational study, I would suggest that the authors re-run the analysis of mortality in hypermagnesaemic patients controlling for raised hydrogen ion (low pH) i.e. exclude those with high H+ (low pH) (and continue to control for renal failure).
Author Response
Response to Reviewer #1
I like this paper a lot
One suggestion for development of the results and discussion sections please:
Hypermagnaesemia is associated with metabolic acidosis. Patients who are initially hypermagnaesemic, may become hypomagnesaemic following correction of acidosis (similar to K+ and DKA). The inverse is also true and may carry more adverse prognostic significance. Therefore, hypermagnesemia in the context of metabolic acidosis may mask a magnesium deficient state. Hence, these fluxes may reflect the severity of metabolic derangement and explain the higher mortality among these patients.
i.e. serum Mg should be interpreted in the context of acid base balance as expressed in pH or H+.
In order for this to be a comprehensive retrospective observational study, I would suggest that the authors re-run the analysis of mortality in hypermagnesaemic patients controlling for raised hydrogen ion (low pH) i.e. exclude those with high H+ (low pH) (and continue to control for renal failure).
Response: We thank you for reviewing our manuscript and for your critical evaluation. We agree with the reviewer’s important comment. The following statements have been added to the result
“The association persisted after patients with in-hospital pH <7.35 or >7.45 were excluded from analysis. In this sensitivity analysis, hypomagnesemia only, hypermagnesemia only, and both hypomagnesemia and hypermagnesemia were significantly associated with increased in-hospital mortality with adjusted odds ratios of 1.58 (95% CI 1.07-2.33), 3.18 (95% CI 2.18-4.64), and 2.56 (1.04-6.32), respectively.”
The following statements have been added to the discussion.
“Our study showed that the incidence of hospital-acquired hypomagnesemia only was 15%, hospital-acquired hypermagnesemia only was 10%, and both hospital-acquired hypomagnesemia and hypermagnesemia was 3%. Patients who developed hospital-acquired hypomagnesemia and hypermagnesemia were associated with higher in-hospital mortality compared to patients who did not. The association also persisted in the sensitivity analysis excluding patients with in-hospital pH <7.35 or >7.45.”
All authors thank the Editors and reviewers for their valuable suggestions. The manuscript has been improved considerably by the suggested revisions!

Reviewer 2 Report
The authors tried to report the incidence of hospital-acquired dysmagnesemia and its association with in-hospital mortality. Their work deserves praise. However, the Discussion part is insufficient. They also need to consider the aspect of how dysmagnesemia affects the cardiovascular system.
You could refer the 2 papers below:
・ATVB 2017;37(8):1431-1445. ・Medicine 2019;98(38):e17069.
Author Response
Response to Reviewer #2
The authors tried to report the incidence of hospital-acquired dysmagnesemia and its association with in-hospital mortality. Their work deserves praise. However, the Discussion part is insufficient. They also need to consider the aspect of how dysmagnesemia affects the cardiovascular system.
You could refer the 2 papers below:
・ATVB 2017;37(8):1431-1445. ・Medicine 2019;98(38):e17069.
Response: We thank you for reviewing our manuscript and for your critical evaluation. We agree with the reviewer’s important comment. The following statements have been added to discussion to describe the role of magnesium in cardiovascular disease as suggested. We have also found suggested references very helpful and have used as new reference (27) and reference (28). The following text in bold has been added in discussion.
“Hypomagnesemia can lead to cardiac arrhythmias by impairing activity of Na/K ATPase and decreasing the negative resting membrane potential. Hypomagnesemia is associated with higher cardiovascular disease as magnesium has a protective role for vascular calcification. Magnesium might bind with phosphate, thereby interfere with calcium phosphate deposition in vessel wall. Additionally, hypomagnesemia can precipitate acute ischemia via endothelial dysfunction and hypercoagulability.”
- Ter Braake AD, Shanahan CM, de Baaij JHF. Magnesium Counteracts Vascular Calcification: Passive Interference or Active Modulation? Arterioscler Thromb Vasc Biol. 2017 Aug;37(8):1431-45.
- Nishihara T, Yamamoto E, Sueta D, Fujisue K, Usuku H, Oike F, et al. Clinical significance of serum magnesium levels in patients with heart failure with preserved ejection fraction. Medicine (Baltimore). 2019 Sep;98(38):e17069.
All authors thank the Editors and reviewers for their valuable suggestions. The manuscript has been improved considerably by the suggested revisions!

Round 2
Reviewer 2 Report
None